# SERS Studies of Adsorption on Gold Surfaces of Mononucleotides with Attached Hexanethiol Moiety: Comparison with Selected Single-Stranded Thiolated DNA Fragments

**DOI:** 10.3390/molecules24213921

**Published:** 2019-10-30

**Authors:** Edyta Pyrak, Aleksandra Jaworska, Andrzej Kudelski

**Affiliations:** 1Faculty of Chemistry, University of Warsaw, 1 Pasteur St., 02-093 Warsaw, Poland; epyrak@chem.uw.edu.pl; 2Nencki Institute of Experimental Biology of Polish Academy of Sciences, 3 Pasteur St., 02-093 Warsaw, Poland

**Keywords:** Raman spectroscopy, surface-enhanced Raman spectroscopy, SERS, alkanethiols, mononucleotides, DNA

## Abstract

The attachment of DNA strands to gold surfaces is performed in many devices, such as various DNA sensors. One of the standard methods used to immobilize DNA on gold surfaces involves two steps: the attachment of a thiol linker group (usually in the form of alkanethiol moiety) to the DNA strand, and the chemical reaction between the thiol-terminated DNA and the gold surface. Since thiols react chemically with the surface of gold substrates, forming very stable Au–S bonds, it is often assumed that the chemisorption on the gold surface of nucleotides with an attached thiol linker group leads to the formation of an order layer with the linking moieties relatively densely packed on the gold surface. In this contribution we show that chemisorption of thiolated mononucleotides does not occur according to this model. For example, the thiolated mononucleotide containing adenine strongly interacts with the gold surface via the adenine moiety. Moreover, bonding of the mononucleotide containing adenine to the gold surface is relatively similar to the bonding of adenine, and the main difference is that the adenine interacts with the gold surface mainly through the pyrimidine ring, while for adenine mononucleotide interaction via the imidazole ring also significantly contributes to the total bonding. A similar effect was observed for the mononucleotide containing cytosine, and the main difference between the interaction with the gold surface of cytosine and cytosine mononucleotide is that mononucleotide containing cytosine interacts with the gold surface to a significantly larger extend via the carboxylic group of the base. We also show that the structure of the layer formed on the gold surface by the thiolated mononucleotides may be significantly different than the structure of the layer formed by thiolated single-stranded DNA containing even as few as two bases.

## 1. Introduction

The attachment of various fragments of DNA to a gold surface is of considerable practical importance, because layers of DNA immobilized on a gold surface are often applied as “a crucial working part” in many DNA sensors. It is worth mentioning that even when different tools (for example, spectroscopic or electrochemical techniques) are used to obtain the analytical signal, very similar fragments of immobilized DNA may be applied to construct such different sensors. One of the standard methods used to attach DNA to a metal, most commonly a gold surface, involves two steps: the attachment of a thiol linker group (usually in the form of an alkanethiol moiety, for example −(CH_2_)_6_−SH or −(CH_2_)_3_−SH), and the chemical reaction between the thiol-terminated DNA and the gold surface. It is well known that thiols (and also dialkylsulfides or dialkyldisulifides [1,2]) react chemically with gold forming a very stable gold-sulfur bond [3,4,5]. The strength of this interaction is very large, the bonds formed during this process are stable, and therefore, alkanethiols form dense monolayers on the gold surface. For this reason, thiolated DNA is widely used to immobilize selected fragments of DNA strands on gold [2,3,4,5]. To illustrate the possible applications of monolayers formed by thiolated DNA on metal surfaces, we have listed below several examples of biosensors utilizing such structures. We have listed only sensors for which the analytical signal is obtained by measuring the surface-enhanced Raman scattering (SERS) signal. 

(i) Thiolated single-stranded capture DNA is immobilized on a roughened metal surface. This structure is then incubated with the target DNA and the reporter DNA tagged with what is known as a Raman reporter (a compound with an exceptionally large cross-section for Raman scattering). In the next step, the Raman spectrum of the whole system is recorded. Where a strong signal is obtained from the Raman reporter, this indicates the presence of the target DNA in the analyzed sample [6,7,8]. A diagram of such a DNA sensor is shown in Figure 1.

(ii) An approach similar to the one described above, but instead of using “reporter DNA tagged with a Raman reporter”, the reporter DNA is tagged with plasmonic nanoparticles with attached molecules of the Raman reporter [9,10,11]. As in the example described above, in the next step, the Raman spectrum of the whole system is recorded, and where a strong signal is obtained from the Raman reporter, this indicates the presence of the target DNA in the analyzed sample.

(iii) Thiolated single-stranded DNA with one end tagged with a Raman reporter is covalently bonded to a SERS-active plasmonic structure. The immobilized DNA forms a hairpin chain, and the Raman reporter is located in close proximity to the plasmonic structure. Such a system generates a very strong SERS spectrum [12,13,14]. In the presence of the specific target DNA, hybridization between the target and the probe DNA disrupts the stem-loop configuration and spatially separates the Raman reporter from the surface of the plasmonic structure, which in turn causes a decrease in the measured SERS signal of the Raman reporter.

(iv) An approach similar to that described in point (iii), but where part of the chain of the immobilized DNA is double-stranded, which prevents the formation of a stem-loop configuration [15,16,17]. In the presence of the specific target DNA, the DNA chain that prevented the formation of the stem-loop configuration is removed (due to hybridization with the target DNA), a hairpin structure is formed, the Raman reporter is moved to within the close proximity of the plasmonic structure, and a strong SERS signal is recorded, which indicates the presence of the target DNA.

(v) Thiolated single-stranded DNA is attached to plasmonic nanoparticles or plasmonic and magnetic nanoparticles [18,19,20]. Due to hybridization, in the presence of the target DNA nanoparticles form agglomerates, which can be easily detected by the increase in the intensity of the measured SERS signal (in the case of plasmonic agglomerates) or by the formation of the plasmonic-magnetic agglomerates, which can be concentrated by the magnetic field. The formation of the agglomerates indicates the presence of the target DNA.

(vi) Thiolated single-stranded DNA is immobilized on a gold surface and the sample is incubated with the analyzed DNA. The presence of the target DNA induces hybridization, which causes a change in the conformation of the linking moiety via which the captured DNA was attached to the gold surface [21]. The change in the structure of the linking moiety is indicated by a characteristic change in the measured SERS spectrum. This experimental procedure is much easier than applying Raman reporters, although the cross section for the Raman scattering of the linking moiety is much lower than when a dye is used.

As illustrated by the above examples, thiolated DNA strands are often immobilized on a gold surface via a chemisorption process, and the systems obtained are very often used in sensors for detecting specific DNA sequences. As the chemisorption of alkanethiols on the gold surface leads to the formation of regular, dense monolayers [22,23], a similar mechanism is attributed to the formation of monolayers by nucleotides with an attached alkanethiol moiety; it is suggested that molecules of thiolated DNA form dense, ordered layers while attached to the gold surface via the Au−S bond [2,3,4,5]. In this contribution we analyze the adsorption on a gold surface of thiolated mononucleotides containing adenine (A), thymine (T), cytosine (C) and guanine (G; see Figure 2). We show that the adsorption of these thiolated mononucleotides is significantly more complex than one might expect from the model presented above (for example, the thiolated mononucleotide containing adenine interacts strongly with the gold surface via the adenine moiety). Surprisingly, for single-stranded DNA containing even just two bases (including adenine), such a strong direct interaction of the adenine moiety with the gold surface has not been observed.

## 2. Results and Discussion

Before recording SERS spectra, electronic absorption spectra were recorded to verify the possibility of nanoparticles’ aggregation. It turned out that maximum of absorbance was observed at 530 nm for all 19 samples including bare nanoparticles (See Appendix A). This observation confirms lack of aggregation of nanoparticles after addition of any DNA nitrogenous bases or nucleotides. 

### 2.1. Adsorption of Different Thiolated Mononucleotides

As mentioned in the experimental section, thiolated mononucleotides were adsorbed on the surface of gold nanoparticles. The SERS spectra of thiolated mononucleotides were compared with the SERS spectra of adenine, cytosine, guanine and thymine, which were also adsorbed on the surface of gold nanoparticles (to facilitate comparison, these spectra are presented on the same plots as the spectra of the respective mononucleotides). Moreover, since the hexanethiol moiety was attached to the mononucleotide as the linking group, we also recorded the SERS spectrum of the 1-hexanethiol adsorbed on the surface of the gold nanoparticles (see Figure 3). The assignment of the most intensive SERS bands of 1-hexanethiol was as follows: the bands at 803 and 862 cm^−1^ were due to the CH_2_ rocking vibration, those at 896, 969 and 1017 cm^−1^ were due to the CH_3_ rocking vibration, those at 1066, 1118 and 1203 cm^−1^ were due to C−C cm^−1^ stretching vibration and the one at 1306 cm^−1^ was due to the CH_2_ wagging vibration [24]. Very characteristic bands in the SERS spectra of chemisorbed alkanethiols were in the region of 600–750 cm^−1^. In our case, the band at 649 cm^−1^ was due to the C−S stretching vibration of molecules having the gauche conformation of the S−C−C chain, whereas the band at 714 cm^−1^ was due to the C−S stretching vibration of molecules having the trans conformation of the S−C−C chain [25].

Figure 4 compares the SERS spectrum of thiolated adenine mononucleotide (a) and the SERS spectrum of adenine (b) adsorbed on the surface of gold nanoparticles. As can be seen, the wavenumbers of the majority of bands in the SERS spectra of thiolated adenine mononucleotide and adenine were usually very similar, for example: the bands at 736 and 737 cm^−1^ (observed for thiolated adenine mononucleotide and adenine, respectively) were assigned to the ring breathing vibration of adenine, the bands at 963 and 968 cm^−1^ were due to the NH_2_ rocking vibration, those at 1339 and 1337 cm^−1^ were due to the C−N stretching vibration of the pyrimidine ring, those at 1376 were due to the C-N stretching vibration of the imidazole ring, those at 1400 and 1399 cm^−1^ were due to the C−N stretching vibration of the pyrimidine ring, those at 1467 and 1455 cm^−1^ were due to the C=N stretching vibrations of the pyrimidine ring and those at 1552 and 1550 cm^−1^ were due to the pyrimidine ring stretching vibration [22,26] (in all cases, the first number was the position of the band for thiolated adenine mononucleotide and the second number was the position of the band for adenine). 

The increase in the Raman intensity due to the SERS effect is complex and it is mainly a result of two effects: (a) an enhancement of the electric field generated by the plasmonic nanostructures and (b) so-called charge transfer enhancement that resembles the ordinary resonance Raman process [23]. In the theory explaining the charge transfer enhancement one assumes the hybridization of the molecular orbital of an adsorbed molecule with an orbital from the metal surface. This allows the resonance Raman process to appear, which leads to an increase in the efficiency of the generation of the Raman signal. The charge transfer enhancement was only observed for molecules directly interacting with the metal surface. Therefore, although the charge transfer enhancement was significantly smaller than the electromagnetic enhancement (charge transfer enhancement is typically equal to only about 1 to 2 orders of magnitude [27]) contribution of the charge transfer enhancement made the SERS spectroscopy very surface-sensitive (measured SERS spectra were usually dominated by the vibrations of the molecules interacting directly with the metal surface).

The large similarity of the wavenumbers of the majority of bands in both SERS spectra suggests that the thiolated mononucleotide interacted with the gold surface mainly via the adenine moiety and not via the thiol group or other parts of the mononucleotide molecule. Moreover, both molecules interacted with gold via the amine group, however, adenine interacted stronger through pyrimidine ring, while mononucleotide rotated towards the imidazole ring. Rings in both molecules were oriented more vertically towards surface (Figure 5). This assumption was also strongly supported by the fact that no noticeable bands characteristic for the chemisorbed alkanethiols (for example at 649, 714, 896 or 1118 cm^−1^ see Figure 3) were observed in the SERS spectrum of the adenine mononucleotide. The frequencies of all the observed bands of this mononucleotide and adenine and their assignments are collected in Table 1.

In Figure 6, the SERS spectra of thiolated cytosine mononucleotide and cytosine adsorbed on gold nanoparticles are shown. As in the previously described case of adenine, (see Figure 4), many bands are present in both SERS spectra, for example, the bands at 796 and 800 cm^−1^ due to the ring breathing vibration, the bands at 1281 and 1310 cm^−1^ due to the C−N stretching vibration of the imidazole ring, those at 1417 and 1423 cm^−1^ due to the C−C stretching vibration, those at 1504 and 1511 cm^−1^ due to the NH_2_ deformation vibration and the bands at 1637 and 1644 cm^−1^ due to the C=O stretching vibration [22,26,28] (in all cases, the first number was the position of the band for thiolated cytosine mononucleotide and the second number was the position of the band for cytosine). Similarities in both spectra (though not so large as in the case of adenine mononucleotide and adenine) suggests that the interaction with the metal surface of the thiolated cytosine mononucleotide also took place to a significant extend via the cytosine moiety. However, the bands appearing at ~1200 and 1040 cm^−1^ (assigned to NH_2_ rocking) in the spectrum of cytosine were not present in the spectrum of mononucleotide, suggesting slight rotation of the molecule and stronger interaction via carboxylic group. Pyrimidine rings in both described molecules were vertically tilted towards surface (Figure 7). The frequencies of all the observed bands of cytosine mononucleotide and cytosine and their assignments are presented in Table 1. 

Similar measurements were performed for thiolated mononucleotides containing guanine and thymine—see the spectra shown in Figure 8 and Figure 9, respectively. However, in these cases, the measured spectra of mononucleotides were significantly different than the respective SERS spectra of guanine and thymine adsorbed on the surface of gold nanoparticles, suggesting stronger contribution of the phosphate group, sugar and hexanethiol influencing at least indirectly its orientation on metal surface. For example, in the SERS spectrum of guanine the most intensive bands were at 666 cm^−1^ (due to the imidazole ring breathing vibration), at 1351 cm^−1^ (C−N stretching in the pyrimidine ring) and at 1699 cm^−1^ (C=O stretching mode) suggesting a rather vertical orientation of pyrimidine ring with the C=O group facing the surface [22,26] (see Figure 10). In the SERS spectrum of the thiolated guanine mononucleotide, the most intensive bands were at 1312 cm^−1^ and 1486 cm^−1^ (due to the respectively C−N and C=N stretching vibrations of the imidazole ring) and at 1576 cm^−1^ (due to the pyrimidine ring stretching vibration) [22,26]. Band at 671 cm^−1^ assigned to imidazole ring breathing shifts, decreased in intensity but its half width increased, suggesting that the molecule was more tilted (flat) towards surface (see Figure 10). This means that for this thiolated mononucleotide, the hexanethiol moiety and/or something other than the nucleotide base parts of these molecules must be involved in the interaction—at the very least affecting the orientation versus the metal surface of the guanine and thymine moiety, which was significantly different than the orientation of guanine and thymine adsorbed in these conditions on the gold surface. For the pair thymine—thiolated thymine mononucleotide SERS spectra varied, however, bands were very wide and overlapping with each other. As thymine did not adsorb strongly on the surface, and as a result it could take different orientations on the particles. 

In summary, for thiolated mononucleotides, the kind of the nucleotide base significantly impacted the interaction of the mononucleotide with the metal surface, in the case of adenine, the base–metal interaction was so strong that the addition of sugar and alkanethiol did not affect it, and in contrast, thymine’s weak interaction changed completely when we compared it with thymine mononucleotide.

### 2.2. Adsorption of Thiolated Single Stranded DNA Chains

As shown above, the kind of the nucleotide base that was present in thiolated mononucleotide significantly affected its adsorption on a gold surface. As the strongest interaction between the nucleotide base in the thiolated mononucleotide and the metal surface was observed for adenine, we decided to verify how the presence of adenine influenced the adsorption of more complex DNA fragments. In the first experiments, we used five different thiolated single-strand DNAs containing one adenine at the 3’ end, to one side of which was attached the hexanethiol moiety and to the other side a DNA chain containing various number of cytosines (1, 3, 7, 11 and 15). The SERS spectra of these thiolated DNA chains adsorbed on gold nanoparticles are shown in Figure 11. As can be seen, even using DNA containing just two bases (one adenine and one cytosine) caused a significant decrease in the relative intensity of the strongest bands characteristic for the adenine moiety. This means that the structure of the layer formed on the gold surface by the thiolated adenine mononucleotide was significantly different than the structure of the layer formed by the thiolated single-stranded DNA containing even only one cytosine base in addition to the adenine base at the 3’ end. A similar effect was observed in experiments with thiolated single-stranded adenine-thymine DNA. In these experiments, we used three different thiolated single-stranded DNA containing 1 adenine at the 3’ end, to one side of which was attached the hexanethiol moiety and to the other side a DNA chain containing various number of thymines (3, 7 and 11). As in the previous experiments with DNA composed mainly of cytosines units, the attachment of the DNA chain containing thymines caused a significant decrease in the relative intensity of the bands characteristic for adenine (spectra not shown).

All the experiments described above involved the use of thiolated single-stranded DNA containing adenine at the 3’ end (to this end of the DNA chain the hexanethiol moiety was also attached). To verify whether the position of the adenine moiety in the DNA chain would influence the DNA’s interaction with the gold surface, similar measurements were carried out using three different thiolated single-stranded DNAs containing 16 bases: 15 cytosines and one adenine, although the adenine was located in three different positions: at the 5’ end as the last base in the chain (AC_15_−(CH_2_)_6_−SH), in the middle of the chain between the 7th and 8th cytosines (C_7_AC_8_−(CH_2_)_6_−SH), and at the 3’ end between the hexanethiol moiety and the cytosine chain (C_15_A−(CH_2_)_6_−SH). As can be seen in Figure 12, all the recorded SERS spectra of these DNA chains were practically identical, and very similar to the spectra in Figure 11. The only difference is the band originating from adenine ring breathing at 734 cm^−1^ appeared in the SERS spectrum of DNA where adenine was in close proximity of hexanethiol moiety. This shows that the adenine–gold interaction was strong enough to change the orientation of whole DNA strand and, on the other hand, SERS spectra of longer than a few bases of DNA strands were very similar, indicating similar position on the nanoparticles.

To confirm these observations, we performed FT-IR measurements for all SERS samples (see Appendix A). The spectra were very similar confirming that there were organic molecules attached to nanoparticles.

## 3. Materials and Methods 

Analytical grade NaH_2_PO_4_·2H_2_O, Na_2_HPO_4_·2H_2_O, adenine, cytosine, guanine, thymine and 1-hexanethiol were purchased from Sigma, Poznan, Poland. 95% sulfuric acid and 30% hydrogen peroxide aqueous solution (both compounds were analytical grade) were purchased from Avantor Performance Materials Poland S.A (Gliwice,). Gold colloids with an average size of nanoparticles of 40 nm were purchased from Ted Pella provided by Unimarket, (Poznan, Poland).

All mono- and oligonucleotides used in this work were purified with HPLC (high-performance liquid chromatography) and were purchased from Genomed, Warsaw, Poland. Nucleotides having the following sentences were used (all the listed sentences are from 5′ to 3′):

A-(CH_2_)_6_-SH, C-(CH_2_)_6_-SH, G-(CH_2_)_6_-SH, T-(CH_2_)_6_-SH, CA-(CH_2_)_6_-SH, C_3_A-(CH_2_)_6_-SH, C_7_A-(CH_2_)_6_-SH, C_11_A-(CH_2_)_6_-SH, C_15_A-(CH_2_)_6_-SH, AC_15_-(CH_2_)_6_-SH, C_7_AC_8_-(CH_2_)_6_-SH, T_3_A-(CH_2_)_6_-SH and T_7_A-(CH_2_)_6_-SH, and T_11_A-(CH_2_)_6_-SH.

The water used in all the experiments was purified in a Millipore Milli-Q system (Burlington, MA, USA). All of the nucleotide’s samples were dissolved in PBS (phosphate buffered saline) with a concentration of Na^+^ of 25 mM. The nucleotide solutions were prepared with a concentration of 1 μM. No TCEP was used for dissolving DNA (thiolated oligonucleotides were received) as the S-gold interaction is much stronger than the S–S bond. The aqueous hexanethiol solutions were prepared with a concentration of 1 mM. It was the lowest 1-hexanethiol concentration for which we managed to record good quality SERS spectra.

Glass slides were cut into 0.5 cm × 0.5 cm pieces, stored in a piranha solution (three parts of concentrated sulfuric acid and 1 part of 30% hydrogen peroxide solution) overnight, washed several times with water and dried under argon. Next, a 30 nm layer of gold was vacuum deposited with the use of a Q150T ES Turbo–Pumped Sputter/Carbon Coater. Clean glass slides were used for comparison; however, as the use of gold-covered glass enhances the signal by about 30–40%, we decided to use these substrates for all the experiments.

Two hundred μL of the solution of Au nanoparticles was mixed with 50 μL of the solution of nucleotides. The mixture was stored overnight, and then centrifuged for 10 min at 2000× *g*. The supernatant was removed and 5 μL of the concentrated solution was deposited on Au-covered glass slides and left to dry at room temperature. Two hundred μL of the solution of Au nanoparticles was mixed with 50 μL of the solution of 1 mM 1-hexanethiol and prepared identically as nucleotides’ samples. Final concentration of nucleotides’ in the sample was 0.2 μM, of 1-hexanethiol–0.2 mM, of nanoparticles: 0.12 nM.

The Raman measurements were performed using a DXR Raman spectrometer produced by Thermo Fisher (Waltham, MA, USA). The Raman spectra were collected using 633 nm excitation radiation emitted by a He–Ne laser with the power at the sample equal to 2 mW. Spectra were recorded by placing a sample on the glass slide and using an air objective (10×) with numerical aperture 0.25 and the diameter of laser spot 2.5 μM. At least five measurements were performed for each sample. All the SERS spectra shown in this work are average spectra from at least five spectra collected at different points of the gold-covered glass slide. The difference in the intensities of the spectra collected from a single sample was below 5% (See Appendix A). 

The absorption spectra were recorded with a UV–Vis-NIR Perkin Elmer spectrophotometer (model Lambda 650, Waltham, MA, USA) in the range of 450–650 nm with a resolution of 2 nm. To place a sample, plastic cells of 1 cm were used.

Infrared spectra were recorded using the Nicolet iS50 FT-IR spectrophotometer from Thermo Scientific. Due to the hydrogel nature of PEDOT–PSS composite IR experiments were carried out using the smart iTR^TM^ Attenuated Total Reflectance Sampling Accessory with the diamond crystal. The spectral resolution was 4 cm^−1^, and typically, 32 scans were averaged for a single spectrum. All IR spectra were preprocessed with atmospheric correction and spectra smoothing (25 points). For FT-IR spectra 2 μL of the samples prepared as for SERS measurements were placed on a diamond crystal and left to dry.

## 4. Conclusions

In this contribution, the interaction of several thiolated mononucleotides and DNA strands with gold nanoparticles were analyzed. Usually, it was believed that such molecules are chemisorbed on the gold surface forming dense, homogenous layers oriented perpendicularly (or only slightly tilted) to the surface. We found that, for mononucleotides, this model did not work, because the direct interaction between the bases and the gold surface might be deduced from the measured SERS spectra. For example, in the case of adenine and thiolated adenine mononucleotide, despite theoretically stronger interaction via sulphur than adenine moiety, SERS spectra were very similar suggesting that alkanethiol linker was further from the surface comparing to the adenine. In contrary, thiolated guanine mononucleotide could interact stronger through the other parts of the molecule than guanine itself. Moreover, even subtle changes in DNA sequence as the position of one base out of 15 could influence its orientation with gold. This observation was indirectly consistent with the literature as we could draw conclusions that for each precise DNA sequence it was crucial to optimize experimental conditions, because with the use of random nanoparticles it could interact with the surface in a very unpredictable way. In our opinion, it should not be stated that thiolated single stranded DNA is oriented vertically on the surface and creates sell assembly monolayer, but rather that for each DNA strand we should be able to find such experimental conditions to make it possible. As in the literature pH, temperature and other factors that might influence the reactions rarely repeat, we could assume that it is not trivial and requires much attention.

## Figures and Tables

**Figure 1 molecules-24-03921-f001:**
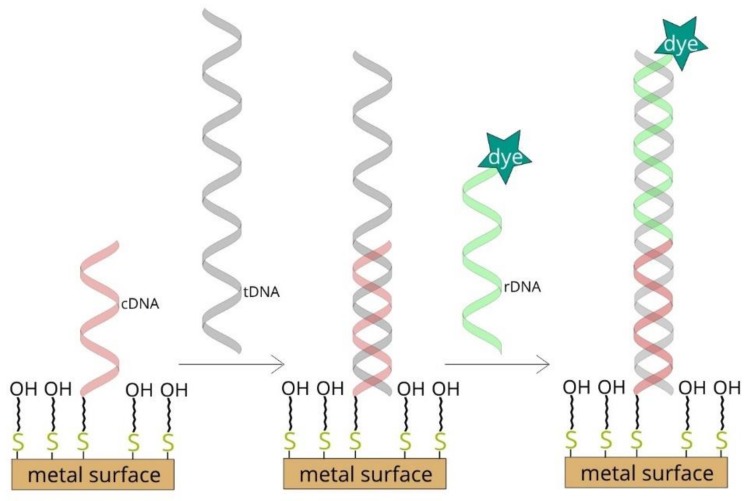
Diagram of surface-enhanced Raman scattering (SERS)-based DNA sensor.

**Figure 2 molecules-24-03921-f002:**
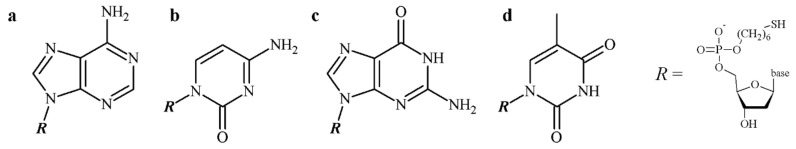
The structure of thiolated mononucleotides of: (**a**) adenine, (**b**) cytosine, (**c**) guanine and (**d**) thymine.

**Figure 3 molecules-24-03921-f003:**
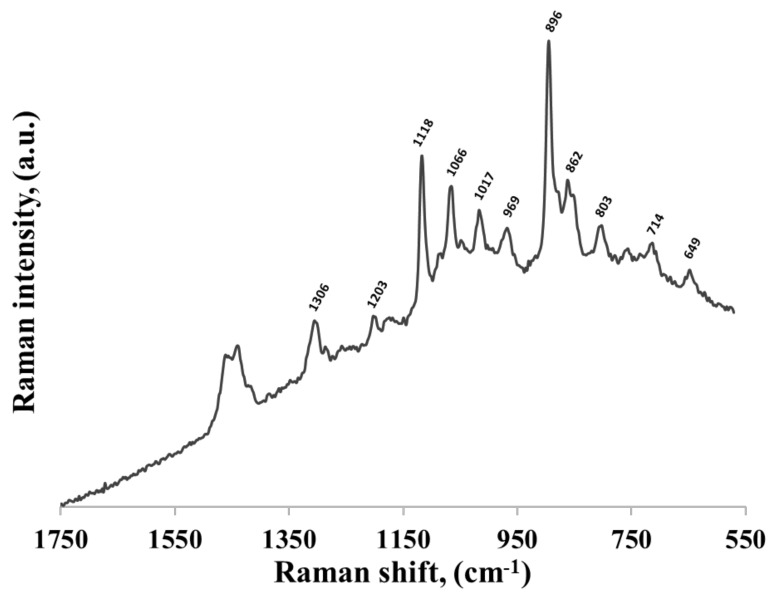
SERS spectrum of 1-hexanethiol adsorbed on the surface of gold nanoparticles.

**Figure 4 molecules-24-03921-f004:**
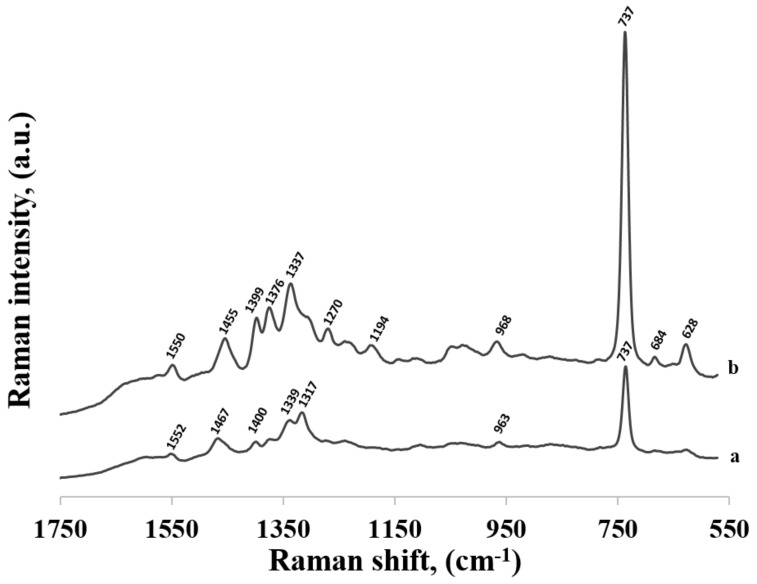
SERS spectra of: (a) thiolated adenine mononucleotide and (b) adenine adsorbed on gold nanoparticles.

**Figure 5 molecules-24-03921-f005:**
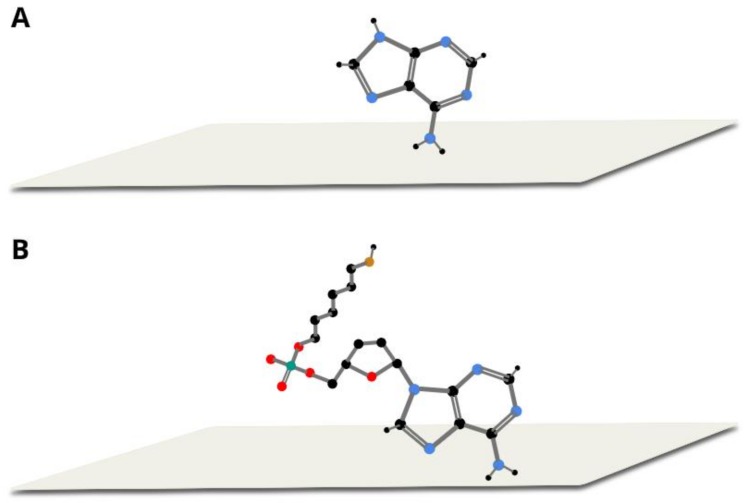
Proposed orientation of adenine (**A**) and thiolated adenine mononucleotide (**B**) on gold colloidal nanoparticles.

**Figure 6 molecules-24-03921-f006:**
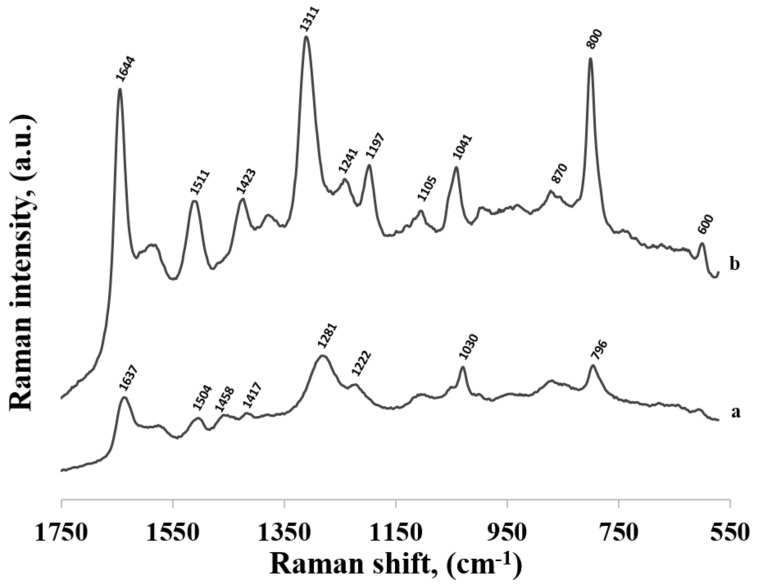
SERS spectra of: (a) thiolated cytosine mononucleotide and (b) cytosine adsorbed on gold nanoparticles.

**Figure 7 molecules-24-03921-f007:**
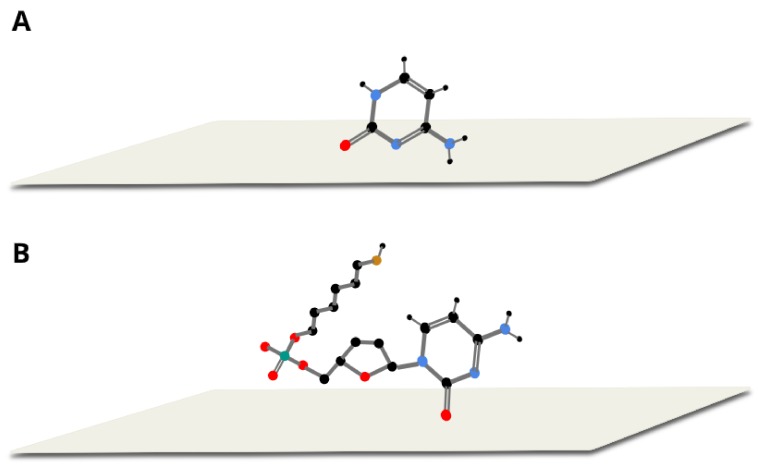
Proposed orientation of cytosine (**A**) and thiolated cytosine mononucleotide (**B**) on gold colloidal nanoparticles.

**Figure 8 molecules-24-03921-f008:**
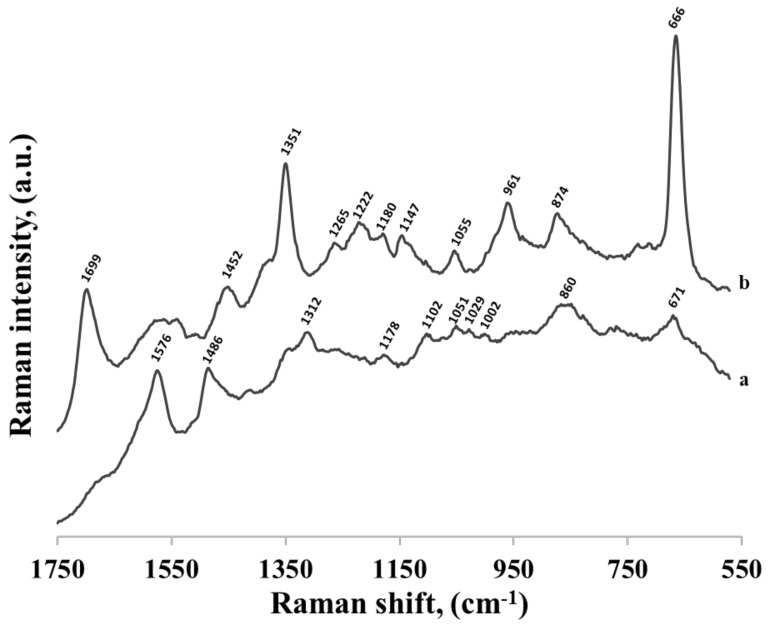
SERS spectra of: (a) thiolated guanine mononucleotide and (b) guanine adsorbed on gold nanoparticles.

**Figure 9 molecules-24-03921-f009:**
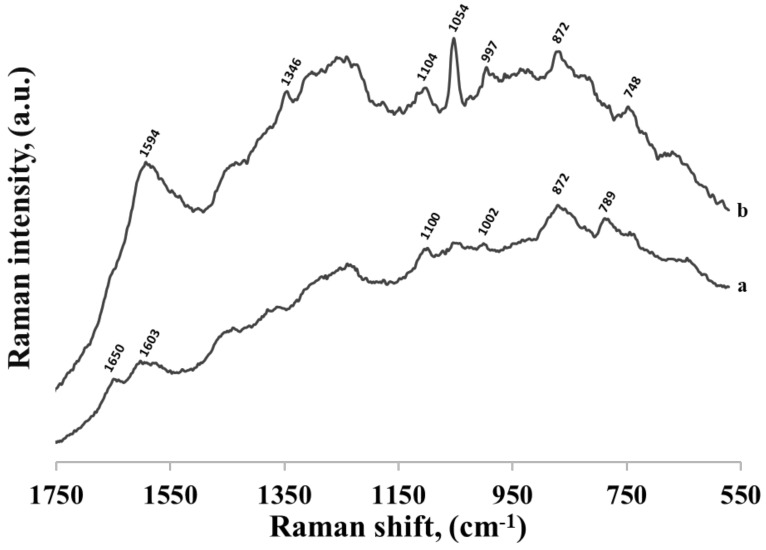
SERS spectra of: (a) thiolated thymine mononucleotide and (b) thymine adsorbed on gold nanoparticles.

**Figure 10 molecules-24-03921-f010:**
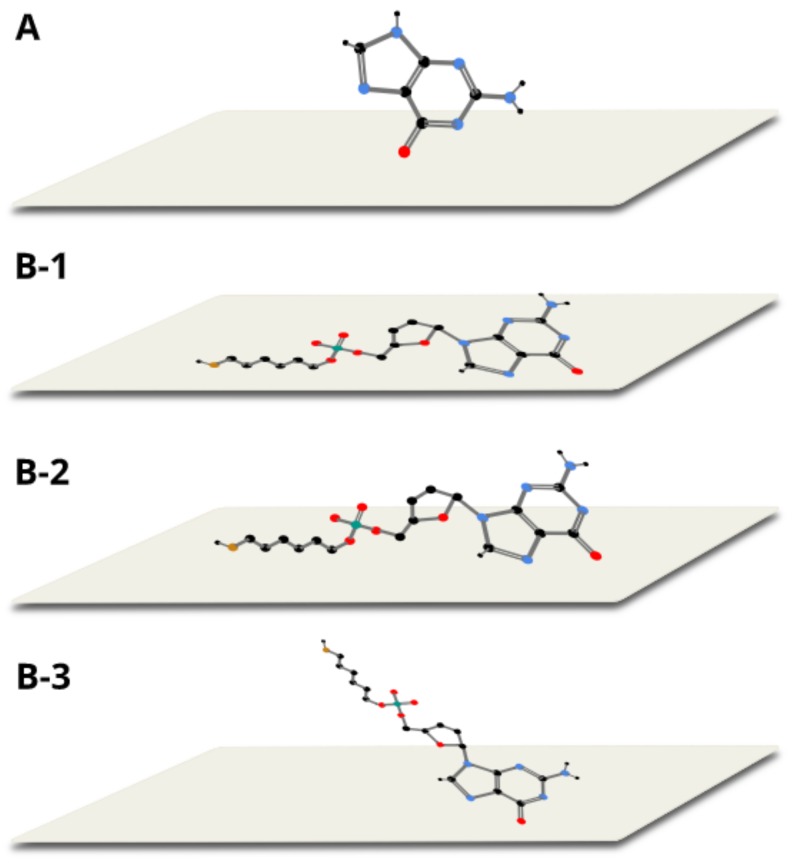
Proposed orientation of guanine (**A**) and thiolated guanine mononucleotide (**B 1–3**) on gold colloidal nanoparticles.

**Figure 11 molecules-24-03921-f011:**
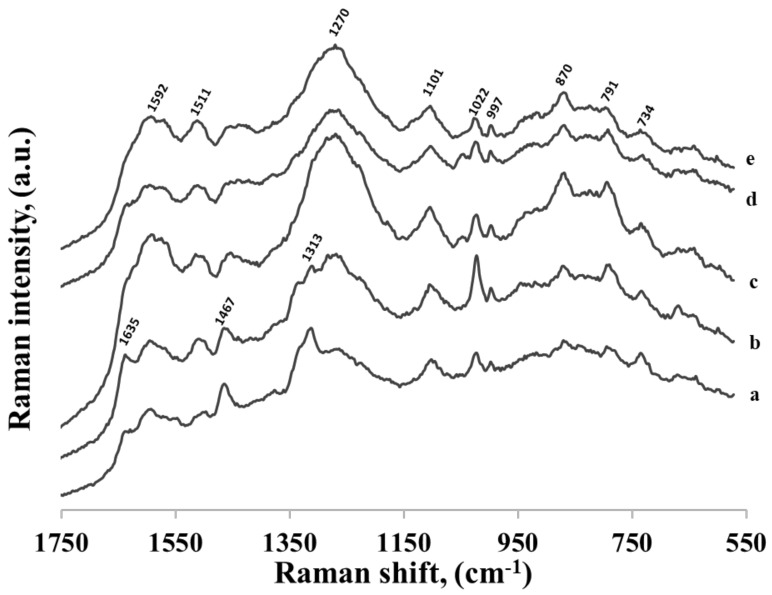
SERS spectra of five different thiolated single-stranded DNA: (a) CA, (b) C_3_A, (c) C_7_A, (d) C_11_A and (e) C_15_A. Adenine is always at the 3’ end, and to this end of the DNA chain hexanethiol moiety is also attached. The nucleotides were adsorbed on gold nanoparticles.

**Figure 12 molecules-24-03921-f012:**
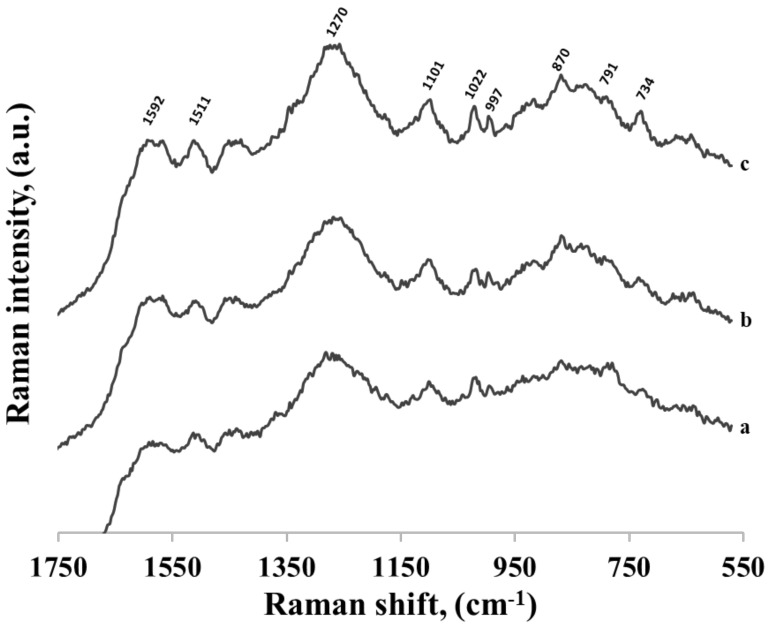
SERS spectra of three different thiolated single-stranded DNA adsorbed on gold nanoparticles. Every DNA contains 16 bases: 15 cytosines and one adenine, however the adenine was located in three different positions: (a) at the 5’ end as the last base in the chain, (b) in the middle of the chain between the 7th and 8th cytosines and (c) at the 3’ end between the hexanethiol moiety and the cytosine chain.

**Table 1 molecules-24-03921-t001:** Assignments of the most intensive Raman bands appearing in the SERS spectra of adenine, cytosine, guanine and thymine and their respective thiolated mononucleotides. The most intensive background bands of gold sols are also presented.

A-SH	Adenine	C-SH	Cytosine	G-SH	Guanine	T-SH	Thymine	Assignments [22,26,28,29,30]
		1637	1644		1699	1650		C=O stretching
						1603	1594	*nanoparticles*
1552	1550			1576				ring stretching (Py)
		1504	1511					NH_2_ deformation
				1486				C=N stretching (Im)
1467	1455	1458			1452			C=N stretching (Py)
		1417	1423					C–C stretching
1400	1399							C–N stretching (Py)
1376	1376							C_6_–N_1_ stretching (Py)
1339	1337				1351		1346	C–N stretching (Py)
1317	1270	1281	1311	1312	1265			C–N stretching (Im)
			1241		1222			C_5_–C_6_ stretching
		1222						ring-CH_3_ stretching
	1194		1197	1178	1180			NH_2_ rocking
					1147			NH_2_ rocking
		1105	1105	1102		1100	1104	*nanoparticles*
				1051	1055	1054	1054	C–C stretching
			1041					NH_2_ rocking
		1030		1029				N-sugar stretching
				1002		1002	997	N–H wagging
963	968				961			NH_2_ rocking
		870	870	860	874	872	872	*nanoparticles*
736	737	796	800			789	748	ring breathing (Py)
	684			671	666			ring breathing (Im)
	628		600					ring deformation (Py)

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
