# Peer review of "SERS Studies of Adsorption on Gold Surfaces of Mononucleotides with Attached Hexanethiol Moiety: Comparison with Selected Single-Stranded Thiolated DNA Fragments"

_molecules, 2019, doi:10.3390/molecules24213921_

Round 1
Reviewer 1 Report
The manuscript "SERS studies of adsorption on gold surfaces of mononucleotides with attached hexanethiol moiety: comparison with selected single-stranded thiolated DNA fragments", written by Pyrak et al. studies an interaction between gold substrate and mononucleotides with 1-hexanethiol. The work presents interesting results uncovering mechanisms behind interactions of hexanethiol-functionalized DNA fragments with a surface of gold nanoparticles, where this combination is frequently used in the development of various nanosensors. However, I do have following comments:
Line 295: authors wrote that the difference between intensities of 5 measured spectra is less than 5%. How was this value calculated and could the authors show representative image of the five spectra collected on one sample? What was the final concentration of mononucleotides and 1-hexanethiol deposited on the surface? What was the size distribution of the Au nanoparticles? This information is usually provided together with the average size. How were the Raman spectra processed? Was there performed any background correction? Were spectra in the fugues shifted? It the Y axis common fo all shown spectra?Author Response
Please see the attachment.

Reviewer 2 Report
The authors describe interaction of adenine and cytosine single bases and the respective thiolated nucleotides with gold nanoparticles (AuNPs). Actually, they only mention these two bases in the Abstract, but in the text, they reveal they also worked with the other two biologically relevant bases, namely, guanine and thymine. This type of study has great relevance in elucidating modes of action of several important DNA molecular assays utilizing oligo-AuNP probes.
The problem I have with all this study is that is based in the concept (never mentioned, but that can be deduced from the presented data), that chemical groups interacting directly with the surface of AuNPs will have their vibrational modes activated on SERS, and that chemical groups that do not interact directly with the AuNP surface have their vibrational modes deactivated or dampened. This concept is very risky, and I wouldn’t advise the authors to do a straightforward assignment of Raman vibrations to SERS vibrations, as the SERS effect is a very complicated a multifactorial one. Also the athour say that ”SERS spectra of thiolated adenine mononucleotide and adenine” “are very similar”, but looking at Figure 4, I can detect several differences even with the naked eye. How were line energies determined? I don’t see reference to any simulation software. None of these limitations, and not even the basis of their method is explained in the manuscript.
The use of complementary molecular structure techniques, such as NMR or even FTIR, would be very helpful in supporting the conclusions.
The conclusion section is very short, only five lines, and does not explain in a convincing manner how the obtained data support the hypotheses.
From the point of view of the manuscript itself, it would greatly benefit the reader if schematics are presented with the chemical structures and the AuNPs surface, indicating the moieties with preferred interactions with AuNPs.
Reviewer 3 Report
In this paper, author systematically investigates the interaction of thiolated mononucleotides, nucleotides, single-stranded HS-DNA with the surface of gold nanoparticles based on the surface enhanced Raman spectrometer. They proved that the presence of adenine influenced the adsorption and compared adsorption strength and possible orientation with thiolated mononucleotides. SERS analysis showed that that the presence of single adenine base is strong enough to change the orientation of whole DNA strand around the surface of Au nanoparticles. According to the reviewer, the manuscript is important for analyzing SERS spectra in nuclear and DNA strands, which is attractive for researching or reading groups in Analytical Chemistry. Therefore, I recommend making minor modifications to this article before accepting it.
In particular, I have the following concerns, questions and suggestions:
Actually, author mixed solutions of Au nanoparticle and nucleotides, so how author confirmed that nucleotides only interacted with one gold nanoparticle, maybe nucleotides in the gap of gold nanoparticles and interacted with both side. A peak at 620 cm-1 is not assigned in Figure 4b. Noted that the Raman intensity of thiolated bases was weaker than the pure ATCG in the whole spectra (see Figures 4-7). Why? In page 9 row 238, the single-stranded DNA (f) C11A is mentioned in the Figure caption, but it is missing in the curve of Figure 8. I suggest to move the picture in Figure 2 to Figure 1 as sub-picture b). The adsorption of these thiolated single nucleotides is much more complicated than expected by the above model. To make the molecular interaction clearer, a scheme to illustrate the adsorption orientation and atoms of different thiolated mononucleotides, nucleotides, and single stranded DNA chains to the surface of Au nanoparticle are needed. Some analysis parts in the experiment descriptions are not clear. The particles size of Au nanoparticle and the reaction concentrations of Au nanoparticles and nucleotides should be provided in the Experiment section. The synthesis method of Au nanoparticles is also necessary to be involved for readers. The Raman measurement methods through fiber probe (quality) or objective (N.A. value and magnification) should be mentioned.Author Response
Please see the attachment

Round 2
Reviewer 2 Report
The authors have made a good effort to improve the quality of their manuscript, namely by increasing the discussion of the theoretical basis of their work, presenting new spectroscopic evidence for their hypothesis and improving substantially the conclusions section. FTIR spectra should be included in the Supporting Information. The manuscript can be published as is.
Author Response
Thank you for that comment. we will attach the IR spectra in supplementary materials